# Probing the Neutron Skin of Unstable Nuclei with Heavy-Ion Collisions

Junping Yang [1], Xiang Chen [1], Ying Cui [1], Zhuxia Li [1] and Yingxun Zhang [1,2,*]

1 China Institute of Atomic Energy, Beijing 102413, China; cuiying@ciae.ac.cn (Y.C.)
2 Guangxi Key Laboratory of Nuclear Physics and Technology, Guangxi Normal University, Guilin 541004, China
* Correspondence: zhyx@ciae.ac.cn

**Abstract:** To improve the constraints of symmetry energy at subsaturation density, measuring and accumulating more neutron skin data for neutron-rich unstable nuclei is naturally required. Aiming to probe the neutron skin of unstable nuclei by using low-intermediate-energy heavy-ion collisions, we develop a new version of an improved quantum molecular dynamics model, in which the neutron skin of the initial nucleus and the mean-field potential in nucleon propagation are consistently treated. Our calculations show that the three observables, such as the cross-sections of the primary projectile-like residues with $A > 100$ ($\sigma_{A>100}$), the difference of $\sigma_{A>100}$ between $^{132}$Sn + $^{124}$Sn and $^{124}$Sn + $^{124}$Sn systems ($\delta\sigma_{A>100}$), and the neutron-to-proton yield ratio ($R(n/p)$) in the transverse direction, could be used to measure the neutron skin of the unstable nuclei and to constrain the slope of the symmetry energy in the future.

**Keywords:** neutron skin; unstable nuclei; symmetry energy; heavy ion collisions

**PACS:** 21.60.Jz; 21.65.Ef; 24.10.Lx; 25.70.-z

## 1. Introduction

The thickness of the neutron skin of a nucleus is characterized as

$$\Delta r_{np} = \langle r_n^2 \rangle^{1/2} - \langle r_p^2 \rangle^{1/2},\tag{1}$$

which reflects the difference between the root-mean-square (rms) radii of the neutron and proton density distributions in the nucleus and is strongly correlated with the slope of the density dependence of the symmetry energy [1–6]. Thus, accurate measurements of the neutron skin of a nucleus can be used to constrain the symmetry energy at subsaturation density.

For the determination of the neutron skin of a nucleus, measurements of the proton and neutron density distributions are needed. The proton density distribution can be accurately determined by electron elastic scattering experiments or isotope shift measurements [7–9], but the neutron density distribution is difficult to measure accurately. The reason is that the neutron is neutral and interacts mainly with hadronic probes. Therefore, the neutron density is mainly probed by proton elastic scattering [10–16], inelastic $\alpha$ scattering [17,18], coherent pion photoproduction scattering [19], antiprotonic atoms [20–22], and relativistic energy heavy-ion collisions (HICs) [23–26]. Another method to measure neutron density is to use weak electric probes, such as the parity-violating e-A scattering method [27–31] of the parity radius experiment at the Jefferson Laboratory, PREX-I [29], PREX-II [30] and CREX [31], or through coherent elastic neutrino–nucleus scattering [32,33]. All of the above methods are mainly used to measure the neutron skin of stable nuclei on the nuclear chart.

A great deal of theoretical analysis on the neutron skin of stable nuclei has been performed to constrain the symmetry energy at a subsaturation density [4,6,34–42], and the

extrapolated values of the symmetry energy coefficient $S_0$ and the slope of the symmetry energy $L$ are in 29–35 MeV and 5–80 MeV [4,35,38,39,43–57], respectively. However, tension appeared after PREX-II published high-accuracy $^{208}$Pb data since the analysis, with a special class of a relativistic mean-field approach, favors a very stiff symmetry energy, i.e., the constrained value of the symmetry energy coefficient $S_0 = 38.1 \pm 4.7$ MeV and the slope of the symmetry energy $L = 106 \pm 37$ MeV [58] are much larger than previously obtained.

To understand the tension and improve the constraints of the symmetry energy at subsaturation density, two aspects should be investigated. One is to understand the influence of cluster mechanisms in the nucleus [59,60], and we will not touch upon this in this work. Another is to enhance the reliability of constraints by using as much data as possible. The neutron skin data for stable nuclei has been analyzed in Refs. [4,6,34–42] to constrain the symmetry energy. For further constraints, accumulating more neutron skin data of neutron-rich unstable nuclei is necessary, and it naturally requires developing a method for measuring the neutron skin of unstable neutron-rich nuclei.

There have been some efforts to measure the neutron skin or the neutron density distribution for unstable nuclei. For example, total reaction cross-sections [61], cross-sections of the isovector spin-dipole resonances (SDR) excited by the ($^3$He, $t$) charge-exchange reaction [62], the strength of pygmy dipole resonances [63], neutron-removal cross-sections in high-energy nuclear collisions [64], and charged pion multiplicity ($\pi^-/\pi^+$) ratios in peripheral heavy-ion collisions [65,66] have been used or proposed. Among these methods, using heavy-ion collisions is most suitable to measure the neutron skin of a very neutron-rich nucleus since the unstable neutron-rich nuclei in a wide range of isospin asymmetry are mainly produced by the projectile fragmentation mechanism in next-generation isotope facilities [67].

However, the measurement of the neutron skin of unstable nuclei via HICs depends on the transport models. In the pioneer transport model calculations [65,66], the slope of the symmetry energy and the thickness of the neutron skin were treated separately. The separate treatment of the neutron skin in the initial nuclei and the mean-field potential in the nucleon propagation increases the theoretical uncertainties for probing the neutron skin of unstable nuclei. Thus, a consistent treatment of the neutron skin in the initialization and the isospin-dependent mean-field potential in nucleon propagation is highly desired.

In this work, the neutron skin of the initial nucleus is correlated to the mean-field potential more consistently using the same Skyrme energy density functional in the updated version of the improved quantum molecular dynamics model (ImQMD-L) [68]. Based on the updated ImQMD-L model, the effects of neutron skin on the collision of $^{124,132}$Sn + $^{124}$Sn at 200 MeV/u are investigated. Our calculations show that the cross-sections of primary projectile-like residues with $A > 100$ can be used to distinguish the thickness of the neutron skin. Furthermore, the energy spectra of the yield ratios of emitted neutrons to protons are also analyzed, which can be used for a complementary understanding of the neutron skin effects and the reaction mechanism.

## 2. Theoretical Framework

In this part, we briefly introduce the form of the potential energy density in the ImQMD-L model and how we correlate the neutron skin of an initial nucleus to the mean-field potential in nucleon propagation with the same Skyrme energy density functional applied.

### 2.1. Potential Energy Density

In the ImQMD-L model [68], the Skyrme-type nucleonic potential energy density without the spin-orbit term is used,

$$
u(\mathbf{r})_{sky} = \quad \frac{\alpha}{2}\frac{\rho^2}{\rho_0} + \frac{\beta}{\eta+1}\frac{\rho^{\eta+1}}{\rho_0^{\eta}} + \frac{g_{sur}}{2\rho_0}(\nabla\rho)^2 \tag{2}
$$
$$
+ \frac{g_{sur,iso}}{\rho_0}[\nabla(\rho_n - \rho_p)]^2
$$
$$
+ A_{sym}\frac{\rho^2}{\rho_0}\delta^2 + B_{sym}\frac{\rho^{\eta+1}}{\rho_0^{\eta}}\delta^2
$$
$$
+ u_{md}.
$$

Here, $\rho$ is the number density of nucleons, which is the summation of neutron density and proton denisity, i.e., $\rho = \rho_n + \rho_p$. $\delta$ is the isospin asymmetry, which is defined as $\delta = \frac{\rho_n - \rho_p}{\rho_n + \rho_p}$. $\alpha$ is the parameter related to the two-body term, $\beta$ and $\eta$ are related to the three-body term, $g_{sur}$ and $g_{sur,iso}$ are related to the surface terms, and $A_{sym}$ and $B_{sym}$ are the coefficients of the symmetry potential and come from the two- and the three-body interaction terms [69]. $u_{md}$ is the Skyrme-type momentum-dependent energy density functional, and it is obtained based on its interaction form $\delta(\mathbf{r}_1 - \mathbf{r}_2)(\mathbf{p}_1 - \mathbf{p}_2)^2$ [70], i.e.,

$$
u_{md}(\mathbf{r},\{\mathbf{p}_i - \mathbf{p}_j\}) \tag{3}
$$
$$
= \quad C_0 \sum_{ij} \int d^3p\, d^3p'\, f_i(\mathbf{r},\mathbf{p}) f_j(\mathbf{r},\mathbf{p}')(\mathbf{p} - \mathbf{p}')^2 +
$$
$$
D_0 \sum_{ij \in n} \int d^3p\, d^3p'\, f_i(\mathbf{r},\mathbf{p}) f_j(\mathbf{r},\mathbf{p}')(\mathbf{p} - \mathbf{p}')^2 +
$$
$$
D_0 \sum_{ij \in p} \int d^3p\, d^3p'\, f_i(\mathbf{r},\mathbf{p}) f_j(\mathbf{r},\mathbf{p}')(\mathbf{p} - \mathbf{p}')^2.
$$

$C_0$ and $D_0$ are the parameters related to the momentum-dependent interaction. More details about it can be found in Ref. [68].

The parameters in Equations (2) and (3) are obtained from the standard Skyrme interaction parameters as in Refs. [43,71]. The connection between the seven parameters, $\alpha$, $\beta$, $\eta$, $A_{sym}$, $B_{sym}$, $C_0$, and $D_0$, used in the ImQMD-L model and the seven nuclear matter parameters, including the saturation density $\rho_0$, the binding energy at the saturation density $E_0$, the incompressibility $K_0$, the symmetry energy coefficient $S_0$, the slope of the symmetry energy $L$, the isoscalar effective mass $m_s^*$, and the isovector effective mass $m_v^*$, are given in Ref. [43]. In the following calculations, the $g_{sur}$ and $g_{sur,iso}$ are 24.5 and $-4.99$ MeVfm$^2$, respectively. Thus, one can alternatively use $\rho_0$, $E_0$, $K_0$, $S_0$, $L$, $m_s^*$, and $m_v^*$ as input to study the influence of different nuclear matter parameters. In this work, we vary only the $L$ to change the thickness of the neutron skin of the nucleus. All the parameters we used are listed in Table 1.

**Table 1.** The values of the nuclear matter parameters used in the ImQMD-L. $m$ is the free nucleon mass. $m_v^*$, $m_s^*$, and $m$ are in MeV, $\rho_0$ is in fm$^{-3}$, $E_0$, $K_0$, $S_0$, and $L$ are in MeV, and $g_{sur}$ and $g_{sur,iso}$ are in MeVfm$^2$.

| $K_0$ | $S_0$ | $E_0$ | $\rho_0$ | $m_v^*/m$ | $m_s^*/m$ | $g_{sur}$ | $g_{sur,iso}$ | $L$ |
|---|---|---|---|---|---|---|---|---|
| 240 | 30 | −16 | 0.16 | 0.7 | 0.8 | 24.5 | −4.99 | 30, 50, 70, 90, 110 |

### 2.2. Initialization with Neutron Skin

To consistently correlate the neutron skin of initial nuclei with the mean-field potential in nucleon propagation, one has to know the neutron and proton density distributions first and then find a way to approximate the density distributions in the ImQMD-L model.

As discussed in Ref. [68], the Woods–Saxon density profile can be reproduced by sampling the centroids of the wave packets within the hard sphere with a radius that equals the half-density radius of the Woods–Saxon density profile. The relationship of the width of the wave packet $\sigma_r$ and the diffuseness of the nucleus $a$ is $\sigma_r = f(a) = (1.71217 \pm 0.01548)a + (0.01564 \pm 0.01047)$ fm. Thus, a model that can be used to calculate the Woods–Saxon-type density distribution under the same Skyrme energy density functional is needed.

The restricted density variational (RDV) method [72] meets this criteria. In the RDV method, the density distributions

$$\rho_i = \rho_{0i} \frac{1}{1 + \exp(\frac{r - R_i}{a_i})}, i = n, p, \tag{4}$$

are adopted, where $R_p$, $a_p$, $R_n$, and $a_n$ are the radius and diffuseness of the proton and neutron density distributions, respectively. $\rho_{0i}$ is the central density of neutrons or protons in the nucleus. These parameters are obtained by minimizing the total energy of the system, which is given by

$$E = \int \mathcal{H} d^3 r = \int \{ \frac{\hbar^2}{2m} [\tau_n(\mathbf{r}) + \tau_p(\mathbf{r})] + u_{sky} + u_{coul} \} d^3 r, \tag{5}$$

under the condition of the conservation of the number of particles in the system, i.e., $N = \int \rho_n(\mathbf{r}) d^3 \mathbf{r}$ and $Z = \int \rho_p(\mathbf{r}) d^3 \mathbf{r}$. $u_{coul}$ is the Coulomb energy density, and $\tau_n$ and $\tau_p$ are the kinetic energy densities of neutrons and protons, respectively. The kinetic energy density in the RDV method is given by the extended Thomas–Fermi (ETF) approach, which includes all terms up to the second order (ETF2) and fourth order (ETF4), as in Ref. [73]. The same semiclassical expression of the Skyrme energy density functional as in ImQMD-L is used to calculate $u_{sky}$.

In Table 2, we show the RDV results of the binding energy $B$, the diffuseness parameters $a_p$ and $a_n$, the half-density radius $R_p$ and $R_n$, the rms radius for neutrons and protons, and the thickness of the neutron skin, respectively. Five Skyrme parameter sets, which are represented by different $L$ values, are used for varying the thickness of the neutron skin. The upper part is for $^{124}$Sn, and the bottom part is for $^{132}$Sn.

**Table 2.** $a_p$, $R_p$, $a_n$, $R_n$, binding energy $B$, and rms radius of neutron and proton density distributions for $^{124}$Sn and $^{132}$Sn obtained with RDV method. $L$ and $B$ are in MeV; $a_p$, $R_p$, $a_n$, $R_n$, $\langle r_p^2 \rangle^{1/2}$, $\langle r_n^2 \rangle^{1/2}$, and $\triangle r_{np}$ are in fm.

| $^{124}$Sn | | | | | | | | |
|---|---|---|---|---|---|---|---|---|
| $L$ | $B$ | $a_p$ | $R_p$ | $a_n$ | $R_n$ | $\langle r_p^2 \rangle^{1/2}$ | $\langle r_n^2 \rangle^{1/2}$ | $\triangle r_{np}$ |
| 30 | −7.971 | 0.414 | 5.733 | 0.514 | 5.777 | 4.670 | 4.865 | 0.165 |
| 50 | −8.021 | 0.415 | 5.729 | 0.507 | 5.811 | 4.698 | 4.881 | 0.183 |
| 70 | −8.073 | 0.419 | 5.707 | 0.503 | 5.838 | 4.687 | 4.893 | 0.206 |
| 90 | −8.129 | 0.422 | 5.686 | 0.496 | 5.872 | 4.676 | 4.907 | 0.232 |
| 110 | −8.191 | 0.426 | 5.656 | 0.487 | 5.909 | 4.659 | 4.922 | 0.263 |

| $^{132}$Sn | | | | | | | | |
|---|---|---|---|---|---|---|---|---|
| $L$ | $B$ | $a_p$ | $R_p$ | $a_n$ | $R_n$ | $\langle r_p^2 \rangle^{1/2}$ | $\langle r_n^2 \rangle^{1/2}$ | $\triangle r_{np}$ |
| 30 | −7.810 | 0.408 | 5.827 | 0.539 | 5.906 | 4.762 | 4.994 | 0.232 |
| 50 | −7.889 | 0.410 | 5.816 | 0.532 | 5.948 | 4.756 | 5.013 | 0.257 |
| 70 | −7.975 | 0.413 | 5.798 | 0.524 | 5.990 | 4.746 | 5.032 | 0.286 |
| 90 | −8.067 | 0.416 | 5.771 | 0.514 | 6.035 | 4.730 | 5.050 | 0.320 |
| 110 | −8.166 | 0.419 | 5.733 | 0.502 | 6.084 | 4.707 | 5.068 | 0.361 |

Then, we prepare the initialization in the same manner as in Ref. [69], but with different treatments in the following two aspects. The first one is that the centroids of the wave packets for neutrons and protons are sampled within the half-density radius of the neutron and the proton obtained in the RDV method, i.e., $R_n$ and $R_p$. Once the positions of all the nucleons have been determined, the density distribution is obtained. Then, the momenta of the nucleons are sampled according to the local density approach. The second one is that the binding energy of the sampled nucleus falling into the range of $B \pm 0.2$ MeV is also required, where $B$ is obtained by the RDV method.

## 3. Results and Discussions

For peripheral collisions at intermediate-energy HICs, there are two characteristics associated with the neutron skin of the nuclei. One is the size of the projectile-like and target-like residues; another is the isospin content of the nucleons and light particles, which are emitted in the transverse direction. The mechanism of the above two characteristics for peripheral HICs is shown in the schematic diagram in Figure 1. Panel (a) is the initial stage of the reaction, panel (b) is at the reaction stage, and panel (c) is at the later stage of the reaction where the projectile and target like fragments are formed and the light articles are emitted from the neck region. Thus, one can expect that a larger neutron skin could lead to a larger reaction cross-section or larger production cross-sections for projectile/target-like residues, as well as the emission of more neutrons and neutron-rich light particles.

**Projectile**

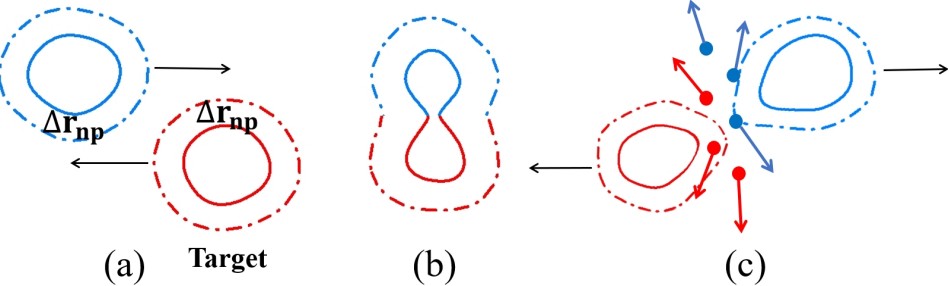

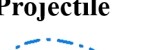

**Figure 1.** Schematic diagram of peripheral heavy-ion collisions. (**a**) Initial stage; (**b**) Reaction stage; (**c**) Light particle emitting stage.

To quantitatively understand the effects of neutron skin on heavy-ion collision observables, we simulate the collisions of $^{124,132}$Sn + $^{124}$Sn at a beam energy of 200 MeV/u and with the impact parameter $b$ ranging from $b = 5$ fm to $b_{max}$ fm with $\Delta b = 0.5$ fm. $b_{max}$ is calculated as

$$b_{max} = R_{proj}^{max} + R_{tar}^{max} + 2.2(a_{proj}^{max} + a_{tar}^{max}). \tag{6}$$

For each impact parameter, 5000 events were simulated. The values of the half-density radius $R_{proj/tar}^{max}$ are the maximum values between the neutron half-density radius and the proton half-density radius of the projectile or target, and the values of $a_{proj/tar}^{max}$ have a similar meaning. The values of $R_{n/p}$ and $a_{n/p}$ are obtained with RDV and are listed in Table 2. The term with $2.2 a^{max}$ in Equation (6) is used to consider the surface thickness of the nucleus.

In Figure 2a, we present $P_{A>100}(b)$, which means the probability of observing a primary heavy residue with mass number $A > 100$ in the forward region, i.e., a projectile-like residue. The red lines are the results of $^{132}$Sn + $^{124}$Sn, and the black lines are the results of $^{124}$Sn + $^{124}$Sn. The $P_{A>100}(b)$ quickly increases from zero to one from semi-peripheral to peripheral collisions, i.e., in the range of $b = 6.2$–9.2 fm ($b = 6.6$–10.0 fm), for $^{132}$Sn + $^{124}$Sn ($^{124}$Sn + $^{124}$Sn). This behavior is determined by the reaction mechanism. When $b < 6$ fm, multifragmentation occurs, and there are no fragments with $A > 100$. At $b > 10$ fm, the distance between the projectile and the target is large enough to produce a heavy projectile-like residue with $A > 100$ in each event, and then $P_{A>100}(b) = 1$. The interesting

point is that curves of $P_{A>100}(b)$ show a sensitivity to the thickness of the neutron skin or to the slope of the symmetry energy. In the range of $b = 6$–10 fm, the larger the values of $L$, the greater the $P_{A>100}$. This is because the calculations with smaller $L$ can reach a higher density in the overlap region, and thus, more compressional energy is stored than for the calculations with large $L$. Thus, the reaction system simulated with smaller $L$ disintegrates into more light fragments compared to the case of the calculations with large $L$. Owing to the conservation of the nucleon number in the reaction system, the production of projectile-like residues with $A > 100$ calculated with larger $L$ is higher than with a smaller $L$. It is consistent with the calculations in Refs. [74,75].

To understand where the projectile-like residues can be detected, we also present the $\theta_{c.m.}$ distribution of projectile-like residues in Figure 2b. The projectile-like residues deflect in a small direction along the beam direction and are distributed within $\theta_{c.m.} < 4°$.

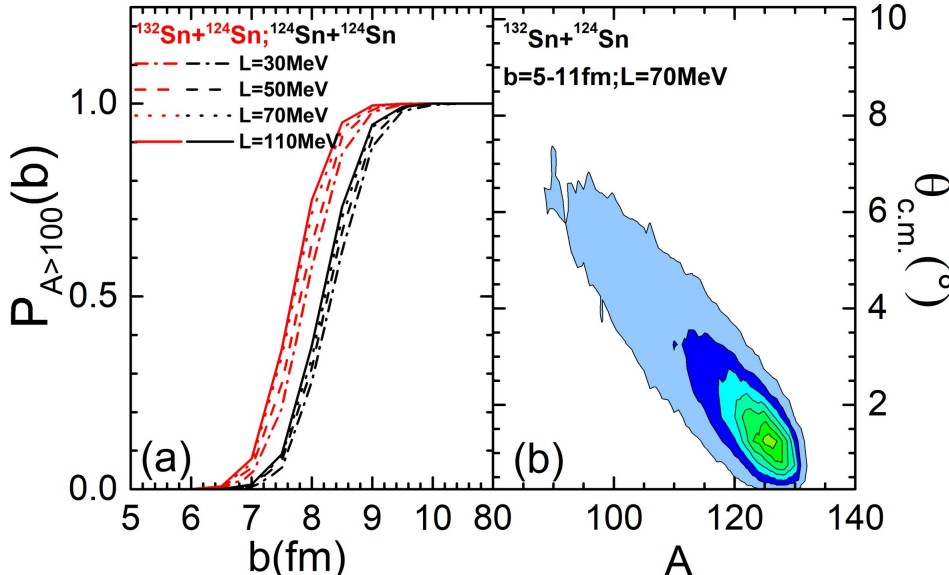

**Figure 2.** Panel (**a**) is the impact parameter dependence of the probability of the primary projectile-like residues with $A > 100$. Panel (**b**) is the $\theta_{c.m.}$ distribution of the projectile-like residues with mass number A.

The cross-section for the projectile-like residues with $A > 100$ is calculated as

$$\sigma_{A>100} = 2\pi \int_0^{b_{max}} P_{A>100}(b) b\, db. \tag{7}$$

In Figure 3, $\sigma_{A>100}$ as a function of the neutron skin thickness of a system, i.e.,

$$\Delta R = \Delta r_{np}^{proj} + \Delta r_{np}^{targ}, \tag{8}$$

is presented. Figure 3a,b are for $^{124}$Sn + $^{124}$Sn and $^{132}$Sn + $^{124}$Sn, respectively. The calculations illustrate that $\sigma_{A>100}$ increases with $\Delta R$ or the slope of the symmetry energy for both systems. For $^{124}$Sn + $^{124}$Sn, $\sigma_{A>100}$ is enhanced by a factor of $\sim$5.2% as $\Delta R$ increases from 0.33 fm to 0.526 fm or as $L$ varies from 30 to 110 MeV. For $^{132}$Sn + $^{124}$Sn, $\sigma_{A>100}$ is enhanced by a factor of $\sim$5.4% as $\Delta R$ increases from 0.397 fm to 0.624 fm. Thus, measuring $\sigma_{A>100}$ could be used to obtain the neutron skin of the nucleus and the slope of the symmetry energy.

Nevertheless, the calculated results of $\sigma_{A>100}$ may be model-dependent or biased. One needs to seek a way to avoid or at least suppress the possible systematic uncertainty caused by the model. Ideally, the difference of $\sigma_{A>100}$ between system $A_{sys}$ and $B_{sys}$, i.e.,

$$\delta\sigma_{A>100} = \sigma_{A>100}(A_{sys}) - \sigma_{A>100}(B_{sys}), \tag{9}$$

where $A_{sys} = {}^{132}\text{Sn} + {}^{124}\text{Sn}$ and $B_{sys} = {}^{124}\text{Sn} + {}^{124}\text{Sn}$, can be used. It is based on a situation in which the systematic uncertainty caused by the same model is similar for the two systems. In Figure 3c, we present $\delta\sigma_{A>100}$ as a function of the difference of the neutron skins between $A_{sys}$ and $B_{sys}$, i.e.,

$$\delta R = \Delta R(A_{sys}) - \Delta R(B_{sys}). \tag{10}$$

Our calculations show that $\delta\sigma_{A>100}$ keeps the sensitivity to $\delta R$ and $L$. Therefore, $\delta\sigma_{A>100}$ could be used to probe the neutron skin of unstable nuclei and constrain the symmetry energy.

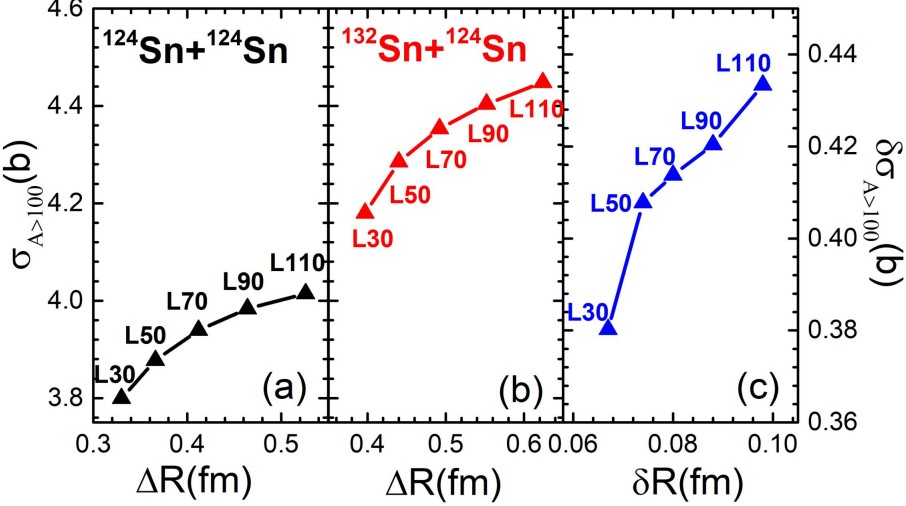

**Figure 3.** Panels (**a**) and (**b**) are the cross-section of the projectile-like residues with $A > 100$ as the function of the neutron skin of the system $\Delta R$ for $^{124}\text{Sn}+^{124}\text{Sn}$ and $^{132}\text{Sn}+^{124}\text{Sn}$, respectively. Panel (**c**) is $\delta\sigma_{A>100}$ as a function of $\delta R$.

As charge size of fragments can be more easily measured than the mass of fragments, we also studied the sensitivity of the cross-section for the projectile-like residues with the charge number $Z > 40$ to $L$. Our calculations show that $\sigma_{Z>40}$ is also sensitive to $L$, but the sensitivity becomes weaker than $\sigma_{A>100}$. This is because the cross-section measured by $\sigma_{Z>40}$ loses the information of the neutron number of fragments, so the sensitivity of $\sigma_{Z>40}$ to $L$ is weaker than $\sigma_{A>100}$.

Next, we analyze the neutron-to-proton yield ratio in the transverse direction. The transverse direction in this work corresponds to $70° < \theta_{c.m.} < 110°$ in the center-of-mass frame. This observable was first proposed to probe the strength of the symmetry potential in Ref. [76] and has been studied extensively for constraining the symmetry energy and effective mass splitting [51,74,77–84].

In this work, $R(n/p)$ is obtained for peripheral collisions, with the impact parameter ranging from 5 fm to 11 fm as

$$R(n/p) = \int_{b=5}^{11} \frac{dY_n(b)}{dE_k} 2\pi b db \Big/ \int_{b=5}^{11} \frac{dY_p(b)}{dE_k} 2\pi b db. \tag{11}$$

This mainly reflects the information of the isospin contents of the overlap region, which is strongly correlated with the thickness of the neutron skin and the slope of the symmetry energy. The calculated results for $R(n/p)$ are presented in Figure 4a,b. The black, red, and blue regions are the results for three values of $\Delta R$, corresponding to $L = 30$, 70, and 110 MeV, respectively. Figure 4a is for $^{124}\text{Sn} + {}^{124}\text{Sn}$, and Figure 4b is for the $^{132}\text{Sn} + {}^{124}\text{Sn}$ system. For the emitted nucleons with kinetic energy $E_k < 80$ MeV, the $R(n/p)$ values are greater for a thin neutron skin than that for a thick neutron skin. This corresponds to the $R(n/p)$ values being greater for the symmetry energy with a small $L$ case than for the symmetry energy with a large $L$. The reason is that the emitted nucleons with lower kinetic energy mainly come from the subsaturation density region, where the symmetry energy obtained

with small $L$ is larger than that with a large $L$. In addition, stronger effects are observed for $^{132}$Sn + $^{124}$Sn than for $^{124}$Sn + $^{124}$Sn.

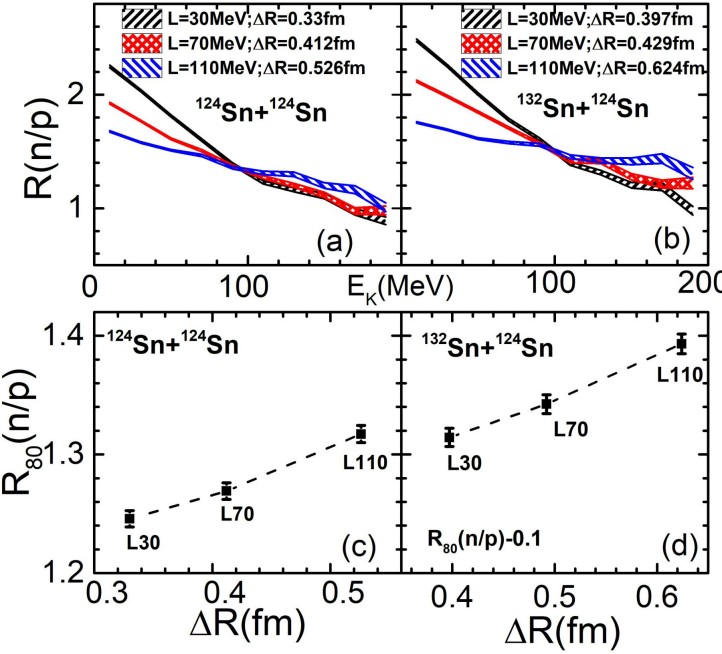

**Figure 4.** Panel (**a**,**b**) show $R(n/p)$ as a function of the kinetic energy of the emitted nucleons; (**c**,**d**) are $R_{80}(n/p)$ as a function of $\Delta R$. $R_{80}(n/p)$ is the ratio of all nucleons above 80 MeV, as shown in Equation (12). The left panels are for $^{124}$Sn + $^{124}$Sn, and the right are for $^{132}$Sn + $^{124}$Sn, respectively. To illustrate the results of two systems in a similar scale, the value of $R_{80}(n/p)$ for $^{132}$Sn + $^{124}$Sn is shifted down by 0.1 in Panel (**d**).

For the emitted nucleons a kinetic energy $E_k \geq 80$ MeV, the $R(n/p)$ values are greater for the symmetry energy with a large $L$ case than for the symmetry energy with a small $L$. The reason is that the emitted nucleons with high kinetic energy mainly come from the high-density region, where the symmetry energy obtained with a large $L$ is larger than with a small $L$. The calculations show that $R(n/p)$ increases with $\Delta R$ but with large uncertainties. To further reduce the uncertainty and distinguish $\Delta R$ with a higher accuracy, the statistics in simulations must be increased. Another way is to integrate the yields of neutrons and protons above 80 MeV and then calculate the neutron-to-proton ratio as

$$R_{80}(n/p) = \frac{\int_{b=5}^{11} Y_n(b, E_k \geq 80) 2\pi b db}{\int_{b=5}^{11} Y_p(b, E_k \geq 80) 2\pi b db}. \tag{12}$$

In Figure 4c,d, we present $R_{80}(n/p)$ as a function of $\Delta R$ for $^{124}$Sn + $^{124}$Sn and $^{132}$Sn + $^{124}$Sn, respectively. According to the absolute statistical uncertainty of $R_{80}(n/p)$ in the current calculation, i.e., $err(R_{80}) = 0.01$, one can distinguish the neutron skin of the unstable nucleus with an accuracy of $\sim$0.02 fm.

## 4. Summary and Outlook

In summary, we consistently correlate the neutron skin of nuclei in the initialization and the isospin dependent mean-field potential in nucleon propagation with the same energy density functional in the improved quantum molecular dynamics for probing the neutron skin of an unstable nucleus. The unstable nucleus of $^{132}$Sn on the target $^{124}$Sn at peripheral collisions and the beam energy of 200 MeV per nucleon is simulated. Our calculations show that the cross-section of projectile-like residues $\sigma_{A>100}$ are correlated with the neutron skin of the system. To avoid the possible systematic deviations from the model, we also construct an observable $\delta\sigma_{A>100} = \sigma_{A>100}(A_{sys}) - \sigma_{A>100}(B_{sys})$, which

reflects the difference of $\sigma_{A>100}$ between two systems $A_{sys} = {}^{132}$Sn + ${}^{124}$Sn and $B_{sys} = {}^{124}$Sn + ${}^{124}$Sn. Our calculations illustrate that $\delta\sigma_{A>100}$ keeps the sensitivity to the thickness of the neutron skin of the unstable nucleus and the slope of the symmetry energy.

In addition, the neutron-to-proton yield ratios, i.e., $R(n/p)$, are also sensitive to the thickness of the neutron skin. In the low-kinetic-energy region, $R(n/p)$ is negatively correlated with the thickness of the neutron skin. In the high-kinetic-energy region, $R(n/p)$ is positively correlated with the thickness of the neutron skin. Thus, the combination analysis on the $\delta\sigma_{A>100}$ and $R(n/p)$ in different kinetic energy regions could improve the reliability and accuracy of the measurements of neutron skins using heavy-ion collisions.

However, one should note that the probe of $\delta\sigma_{A>100}$ requires experimentalists to develop a method for reconstructing primary projectile-like residues from the emitted light particles and cold fragments. Currently, a kinematical focusing method has been developed for the reconstruction of intermediate-mass fragments in Ref. [85]. Even though there will be many difficulties in reconstructing the primary projectile-like residues, it still implies that the observables constructed from primary projectile-like residues may be in practical use in the future, and the probe of $\delta\sigma_{A>100}$ and $R(n/p)$ should be considered in the potential wish list of experimenters.

**Author Contributions:** Conceptualization, Y.Z.; methodology, J.Y., X.C. and Y.Z.; software, Y.Z.; validation, J.Y., X.C., Y.C. and Y.Z.; formal analysis, J.Y. and Y.Z.; investigation, J.Y. and Y.Z.; resources, Y.Z.; data curation, J.Y.; writing—original draft preparation, J.Y. and Y.Z.; writing—review and editing, J.Y., X.C., Y.C., Z.L. and Y.Z.; visualization, J.Y.; supervision, Y.Z.; project administration, Y.Z.; funding acquisition, Y.Z. All authors have read and agreed to the published version of the manuscript.

**Funding:** This work was partly inspired by the transport code comparison project, and it was supported by the National Natural Science Foundation of China Nos. 12275359, 11875323, and 11961141003, the National Key R&D Program of China under Grant No. 2018YFA0404404, the Continuous Basic Scientific Research Project (No. WDJC-2019-13, BJ20002501), and the funding of the China Institute of Atomic Energy (No. YZ222407001301). The work was carried out at the National Supercomputer Center in Tianjin, and the calculations were performed on TianHe-1 (A).

**Data Availability Statement:** The data presented in this study (i.e., data from simulations) are available on request from the corresponding author.

**Acknowledgments:** The authors are thankful for the useful discussion with Z.G. Xiao and W.P. Lin.

**Conflicts of Interest:** The authors declare no conflict of interest.

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
