# Peer review of "Probing the Neutron Skin of Unstable Nuclei with Heavy-Ion Collisions"

_universe, doi:10.3390/universe9050206_

Round 1

Reviewer 1 Report

In this work the authors used a consistent treatment of the neutron skin in the initialization and the isospin dependent mean field potential in nucleon propagation in the updated version of improved quantum molecular dynamics model (ImQMD-L). Their calculations clearly show that the cross sections of the primary projectile-like residues with A > 100, the difference of these cross sections between 132Sn+124Sn and 124Sn+124Sn, and the neutron to proton yield ratio in the transverse direction, could be used to measure the neutron skin of the unstable nuclei and to constrain the slope of the symmetry energy.

The results are solid, clearly presented and explained. The paper contains an extended bibliography on neutron skin studies and their relations to symmetry energy of nuclear matter. I recommend the text for publication in the journal after taking into account the following minor remarks. 

L5: between system 132 Sn+124Sn and 124Sn+124Sn --> between 132 Sn+124Sn and 124Sn+124Sn systems

L12-13: radius of the neutron and proton --> radii of the neutron and proton density distributions in the nucleus

L42: to develope --> to develop 

(Develope is an old British spelling of the word. Develop is a more modern, American way of spelling the word. Moreover, it seems that American English is used in the manuscript.)  

L63: improvde --> improved

Eq.(2): Please define $\delta$ used in this equation.

L107: of neutron or proton --> of neutrons or protons

Caption of Table 1: Please introduce here $m$ as the free (on-shell) nucleon mass

Eq.(5): Please define $u_{coul}$, $\tai_n$ and $\tai_p$

Caption of Table 2: radius for neutron and proton for --> radii of the neutron and proton density distributions in 

L152: b <∼ 6 --> b < 6  or use $\leq$

L153: b >∼ 10 fm  --> b > 10 fm  or use $\geq$

Caption of Fig.4: The ratio $R_{80}(n/p)$ has to be introduced here as the ratio specifically for nucleons below 80 MeV or a reference to Eq.(12) to be given.

L216: $err(R_{80})=0.01$ Is is just a sigma value? Is it relative or absolute uncertainty? Please specify this in the text.

References: 

1) Avoid repeating twice "https://doi.org/" in links of some entries

2) Use abbreviated journal names

3) No need to capitalize the title of [50]. Give the reference to http://dx.doi.org/10.1088/0004-637X/771/1/51 (The Astrophysical Journal) instead of arXiv

4) In [60] give a proper reference to Science journal instead if arXiv

Reviewer 2 Report

The manuscript deals with the effect of neutron skin on the cross section as well as the n/p ratios in peripheral heavy ion collisions with unstable colliding nuclei 132Sn. The authors find that both \sigma(A>100) and R(n/p) are sensitive to the slope parameter of the symmetry energy. The analysis is well done and the paper is well written. 

 Overall, I recommend the manuscript for publication. Some mirror suggestions I would like to see addressed are: (1)  Does the charge size change as the slope parameter L varies? As this can be measured with good precision, some discussion of this would be useful. (2) It would be better to give the relationship between the wave packet and the diffuseness of the nuclei for convenience, then the potential reader does not need to dig into the relevant references. 

Reviewer 3 Report

In the manuscript, the authors propose the use of the data from collisions of Tin 124 and 136 to probe the neutron skin of unstable nuclei. The authors performed theoretical calculations and simulations to illustrate their point. Neutron skin has always been an intriguing research subject in nuclear physics. A successful measurement of the neutron skin offers a chance to measure the density of the entire nucleus. Even though the authors are not the first to propose the use of LHC but none has studied Tin for this matter. Therefore, what’s covered in this manuscript is of significance and novelty. The methods are adequately described, and the results are clearly presented. I recommend its publication in Universe after minor revisions for the following reasons:

1.       The authors need to add a sentence that describes the significance of neutron skin measurements of unstable nuclei in the abstract.

2.       In section 1, the authors need to illustrate the interest regarding unstable nuclei neutron skin measurements. Neutron skin is important and there is already a successful measurement in 2021. Then why should we care about that of unstable nuclei?

3.       “Coherent elastic neutrinos scattering” at line 26 should be coherent elastic neutrino-nucleus scattering.

4.       The authors need to reformat section 3 Results and discussions. In this section, authors talked about the collision simulation performed, which should have been included in section 2. So are the formulas used to calculate the neutron skin. Also please include more details about the simulation performed: what software is used? is post-processing needed? Then the rest of section 3 belongs to “analysis” rather than results and discussions.
